# LEO Precise Orbit Determination with Inter-satellite Links

**Xingxing Li \*, Zihao Jiang, Fujian Ma, Hongbo Lv, Yongqiang Yuan and Xin Li**

School of Geodesy and Geomatics, Wuhan University, 129 Luoyu Road, Wuhan 430079, China

\* Correspondence: xxli@sgg.whu.edu.cn

**Abstract:** Traditional precise orbit determination (POD) for low Earth orbit (LEO) satellites relies on observations from ground stations and onboard receivers. Although the accuracy can reach centimeter level, there are still problems such as insufficient autonomous operation capability. The inter-satellite link (ISL) is a link used for communication between satellites and has a function of dual-way ranging. Numerous studies have shown that observational data using ISLs can be adopted for POD of navigation satellites. In this contribution, we mainly focus on LEO satellites POD with ISLs. First, we design LEO constellations with different numbers of satellites and ISL measurements, based on which the constellations are simulated. Then rough tests of POD using different link topologies are carried out. The results show that in the 60-LEO constellation the average 3-dimensional (3D) orbital errors are 0.112 m using "4-connected" link topology with constant 4 links per satellite and 0.069 m using "all-connected" link topology with theoretically maximum numbers of links. After that, we carry out refined POD experiments with several sets of satellite numbers and different observation accuracy. The results show the higher link ranging accuracy and the more numbers of links bring higher POD precision. POD with ISLs gets bad performance in the case of center of gravity reference when link ranging accuracy is poor and numbers of links are small. When the link accuracy is 40 cm, average 3D orbital errors of 60-LEO constellation are 0.358 m, which can only meet the demand of autonomous navigation. With the constraint of the right ascension of the ascending node (RAAN), POD using ISLs reaches an extremely high precision when adopting a spatial reference provided by navigation satellites. For 120-LEO constellation, the average 3D orbital errors are 0.010 m; for 192-LEO constellation, the errors are 0.006 m.

**Keywords:** precise orbit determination; low Earth orbit satellites; inter-satellite link; spatial reference

## 1. Introduction

The precise orbit determination (POD) of satellite refers to the process of obtaining the accurate state vectors of satellites in a certain epoch by handling satellite orbital measurements with the estimation strategies like least squares or Kalman filtering [1,2]. The measurement technique exploits a decisive role in the accuracy of the POD process, and the selection of the POD method should be adapted to the orbit measuring technology, combined with the actual situation of the satellites and the users' needs [3].

For global navigation satellite system (GNSS), which usually employs medium Earth orbit (MEO, altitude of 2000–20,000 km), geostationary Earth orbit (GEO, altitude of 35,786 km), and inclined geosynchronous orbit (IGSO, altitude of 35,786 km), the main source of orbit measurements are pseudo-range and carrier phase observations of ground-based monitoring stations. With the refinement of force model as well as the improvement of data processing strategy, POD for GNSS satellites can reach an accuracy of centimeter level [4,5]. However, POD for GNSS satellites relies heavily on ground

measurements. Once ground receivers lose communication with the satellites because of crisis situation like equipment downtime, orbit determination becomes a problem.

In order to solve the above problems and further improve the accuracy of orbit determination, some scholars [6–8] have proposed the use of inter-satellite link (ISL) for POD of GNSS satellites. The ISL is a link used for communication between satellites and has a function of ranging. It is a wireless link that can be implemented using a laser link or a microwave link. The microwave link can reach centimeter-class ranging accuracy while the laser link can even reduce the ranging errors to sub-millimeter [9]. Though more precise in ranging, the laser link has higher power consumption as well as longer link acquisition time, so the microwave link is generally used in the constellation. The development trend of microwave ISLs is to use higher frequency bands, because it can reduce the weight and volume of the equipment, and the high frequency has not been occupied in a large amount, which benefits the frequency application and has strong anti-interference performance [10]. The bands that can be used for the ISL are very high frequency (VHF, 30–300 MHz), ultrahigh frequency (UHF, 300–3000 MHz), L (1–2 GHz), S (2–4 GHz), and K-above (Ka, 26.5–40 GHz) bands, while the L-band and S-band are already occupied by the satellite-ground links, so the ISL band is generally selected from the VHF, UHF, and Ka bands [11–13]. Since the ISL technology can obtain high-precision ranging values between satellites, it can be used as an auxiliary means of orbit observing and serve for POD.

The American Global Positioning System (GPS) is the first constellation to successfully use the ISLs to achieve autonomous navigation [14]. In 1998, the United States launched the first Block IIR satellite using the UHF band. Since 2010, the Block IIF series launched into the sky, improving the link structure and function. Currently, the GPS system is preparing to increase the Ka-band ISL equipped on the GPS III satellites. In this way, the master station only needs to establish contact with a satellite to control the entire constellation [15]. The first ISL POD experiment has been carried out when Block IIR satellites carrying ISL payloads entered the orbit and the user ranging error (URE) of the autonomous determined orbits within 75 days is less than 3 m [16]. In order to comprehensively analyze the performance of ISL POD, further researches have been conducted. Data analysis shows that after adding the Block IIR ISL, the 180-day autonomous POD URE is less than 6 m and after adding the Block IIF ISL, the reduction of the original system URE is improved from 40 to 55% [17]. because of the critical technology to be solved, the launch plan of GPS III satellites was postponed until 2017 [18]. Therefore, there are currently no public data on the Ka-band ISL and it is impossible to evaluate its orbit determination performance.

In addition to the GPS, the European Galileo and Russian Globalnaya Navigatsionnaya Sputnikovaya Sistema (GLObal NAvigation Satellite System, GLONASS) are also equipped with ISL payloads. The Galileo system has designed two link schemes of GNSS+ and ADVISE which establish inter-satellite and satellite-to-earth links using transmit/receive antennas with the same frequency band [19]. Through experimental demonstration, when the GNSS+ scheme is adopted in the autonomous POD mode, the numbers of ground stations can be reduced to six while reaching an accuracy of meter level in 14 days. And when using the ADVISE scheme, the orbital accuracy can be improved by 67% [20]. In the future, the Galileo system will introduce the so-called optical-quantum link scheme based on frontier quantum communication technology in substitution of microwave links [21]. In order to solve the limitation of regional monitoring and tracking for GLONASS constellation, Russia has installed the S-band ISL transceiver on the GLONASS-M satellites to verify the function of ISLs [22]. Since the test results were promising, the laser links were used on the newly launched GLONASS-K satellites, and the POD error is less than 50 cm in the radial direction and less than 110 cm in the along-track direction [23].

Chinese new-generation BeiDou Navigation Satellite System (BDS), so-called BDS-3, are equipped with ISL payloads, which are capable of communications between satellites with Ka-band single-frequency pseudo-code ranging measurements [24]. Based on the whole network adjustment algorithm, a centralized kinematics POD algorithm using BDS-3 ISLs has been proposed by Chen et al. [25]. With the simulated full operational capacity (FOC) BDS-3 ISLs data, the average

POD errors obtained under 40 cm measurement noise are 0.80, 1.15, and 1.17 m in radial, along-track, and cross-track directions, respectively. Feng et al. have proposed the joint POD model of ground observations, LEO satellite observations, and BDS-3 (including 3 GEO satellites, 3 IGSO satellites and 24 MEO satellites) ISL measurements based on simulation [26]. Under the ISL accuracy of 10 cm, the average orbital errors of BDS-3 constellation in the radial, along-track, and cross-track directions are 0.003, 0.254, and 0.200 m, respectively. Moreover, the introduction of BDS-3 ISLs shows more improvement in the orbital accuracies than the introduction of LEO satellites, compared to the traditional ground-based POD strategy. With the BDS-3 ISL data being accessible, more scientific researches have been carried out, which are concentrated on BDS-3 autonomous navigation and joint POD of ground observation and ISL data. Ren et al. have compared and analyzed POD results of four experimental BDS-3 satellites with only ISL measurements in 4 days and the prior orbits have been used as constraints and the POD precision with two, three, and four ISLs is better than 6, 3, and 2 m, respectively [27]. Tang et al. have investigated the centralized autonomous orbit determination using ISL observations with the support of ground anchor stations [28]. The results show that the radial overlap differences and satellite laser ranging (SLR) residuals are both less than 15 cm. Xie et al. have evaluated the POD performance of BDS-3 satellites with satellite-ground and inter-satellite observation data [29]. Considering the regional distribution of the BDS-3 ground stations, only six stations within the Chinese border are used and the statistical results show after adding the BDS-3 ISL observations, the 3-dimensional (3D) RMS value of orbit overlap differences is reduced from 85.4 to 14.8 cm with great improvement of 83%.

In summary, the ISL can provide high-precision orbit observation data and can be applied to POD, however, current researches on ISL POD are limited to GNSS satellites. Low Earth orbit (LEO) satellites have lower orbital altitude (500–2000 km) than MEO, IGSO, and GEO satellites, so the orbital measurements of LEO satellites mainly come from downlink data of higher GNSS satellites which are received by onboard receivers. POD for LEO satellites include "two-step" approach (doing POD for GNSS satellites at first with ground measurements and then doing POD for LEO satellites with onboard measurements) [30] and "one-step" approach" (the orbits of LEO and GNSS satellites are recovered in one simultaneous process with both ground and onboard measurements) [31], both of which can reach an accuracy of a few centimeters [32]. Similar to navigation satellites, POD of LEO satellites is highly dependent on the ground or onboard ranging data. Since the data processing strategy is unrelated to orbital altitude, it is confirmed to be feasible to introduce in-orbit ISL measurements for POD of LEO satellites. Nowadays, LEO constellation has become a new hotspot and is vital to the future of satellite industry and scientific researches [33]. Because the acquisition of LEO satellite orbits is a prerequisite for putting them into the market or scientific applications, it is also of great significance to study the POD of LEO satellites based on ISLs. Chinese Hongyan LEO constellation equipped with the ISL payloads is under construction and will complete the deployment of main constellation by 2023 [34]. Undoubtedly, it will facilitate the future research of LEO satellite POD with ISLs.

Our study mainly focuses on preliminary evaluation of the performance of LEO satellite POD with ISL measurements. In the following Section 2, basic concept of ISL is introduced and its observation model is described. Preparation of orbit determination experiment, including the design of LEO constellation, the simulation of ISL measurements, and the selection of link topology, is detailed in Section 3. Besides, our method for POD with ISLs is also introduced. Section 4 presents the result and analysis. Then the shortcomings of the work and an outlook for future research are summarized in Section 5. Finally, Section 6, presents the conclusion of our research.

## 2. ISL Observation Model

In principle, two satellites can establish an ISL as long as they are visible to each other [35]. A pair of satellites (i.e., satellite *A* and *B*) and the Earth are shown in the Figure 1, with O being the center

of the Earth and R being the radius of the Earth. Our definition of "visible" for satellite $A$ and $B$ is formulated as follow:

$$\frac{\left| \overrightarrow{OA} \times \overrightarrow{OB} \right|}{\left| \overrightarrow{OA} - \overrightarrow{OB} \right|} \geq R \tag{1}$$

In other words, satellite $A$ and $B$ are considered to have the capacity of an ISL if Equation (1) can be satisfied. It is worth mentioning that in practical application the direction of satellite antenna also impose restriction on ISL. Taking BDS-3 as an example, a phased array antenna switches the beam direction during different timeslots in order to point to different satellites, and the pointing limit is $(-60°, 60°)$ [36]. Since all our work are based on simulated constellation, we neglect the effect of satellite antenna direction for simplicity.

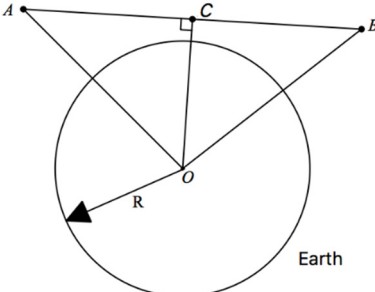

**Figure 1.** A pair of visible satellites.

The ISL ranging is a dual one-way observation mode with a pair of satellites receiving the signal from each other almost simultaneously, and it is only capable of pseudo-code measurements. Figure 2 vividly depicts the ISL mechanism between satellite $A$ and $B$. Given that satellite $B$ receives the pseudo-code measurement $\widetilde{\rho}_A^B(t_A)$ from satellite $A$ at its local time $t_A$ while satellite $A$ receives the pseudo-code measurement $\widetilde{\rho}_B^A(t_B)$ from satellite $B$ at its local time $t_B$ and the forward and backward signal transmitting time is $\Delta t_{AB}$ and $\Delta t_{BA}$, respectively. Then, the dual pseudo-code measurements can be expressed as follows:

$$\widetilde{\rho}_A^B(t_A) = \rho_A^B(t_A + \Delta t_{AB}) + \delta_B(t_A + \Delta t_{AB}) - \delta_A(t_A) + l_B^R - l_A^S + O_{AB} + \varepsilon_{BA} \tag{2}$$

$$\widetilde{\rho}_B^A(t_B) = \rho_B^A(t_B + \Delta t_{BA}) + \delta_A(t_B + \Delta t_{BA}) - \delta_B(t_B) + l_A^R - l_B^S + O_{BA} + \varepsilon_{BA} \tag{3}$$

where $\widetilde{\rho}_A^B(t_A)$ and $\widetilde{\rho}_B^A(t_B)$ are the measured ranges at signal receiving epoch of each satellite and $\rho_A^B = c * \Delta t_{AB}$ and $\rho_B^A = c * \Delta t_{AB}$ ($c$ is the speed of light travelling in vacuum space, which is about 299,792.458 km/s) are the forward and backward signal space propagation (i.e., the real ranges between satellites at their signal receiving epoch), respectively; $\delta_A$ and $\delta_B$ represent the clock bias of each satellite; $l_A^S$ and $l_B^S$ denote the hardware delay of satellite $A$ and $B$ at the sending end while $l_A^R$ and $l_B^R$ at the receiving end; $O_{AB}$ and $O_{BA}$ are phase center offsets (PCOs), phase center variations (PCVs), relativistic effects, and some other corrections; $\varepsilon_{AB}$ and $\varepsilon_{BA}$ are the corresponding measurement noise. It is worth noting that the abovementioned ISL model is defined for GNSS constellations. Since there are currently no LEO satellites with ISL loads, it is difficult to analyze the error factors affecting the inter-satellite ranging under LEO conditions. So we use the GNSS ISL observation model as substitution for the LEO ISL.

The forward and backward ISL observations are nearly at the same time but still have differences. In the BDS-3 ISL observations, the time difference is less than 3 s [28]. So at first, what we need is to transform the dual measurements to the same time $t_0$ with the auxiliaries of broadcast orbits and clock biases. This procedure is defined as epoch naturalization, during which all the hardware delays are considered to be constant.

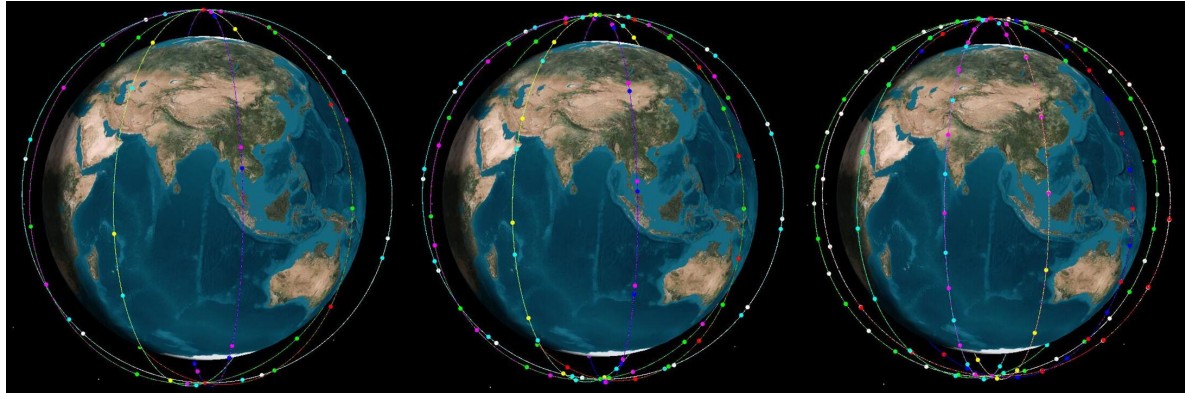

**Figure 2.** Configuration of 60-LEO (**left**), 120-LEO (**middle**), and 192-LEO (**right**) constellation.

After epoch naturalization and corrections of PCOs, PCVs, and relativistic effects, we can obtain the pure dual pseudo-code measurements at the same time, which are expressed as follows:

$$\widetilde{\rho}_A^B(t_0) = \widetilde{\rho}_A^B(t_A) + \Delta\rho_A^B = \rho_{AB}(t_0) + \delta_B(t_0) - \delta_A(t_0) + l_B^R - l_A^S + \varepsilon_{AB} \tag{4}$$

$$\widetilde{\rho}_B^A(t_0) = \widetilde{\rho}_B^A(t_B) + \Delta\rho_B^A = \rho_{AB}(t_0) + \delta_A(t_0) - \delta_B(t_0) + l_A^R - l_B^S + \varepsilon_{BA} \tag{5}$$

where $\rho_{AB}(t_0)$ is the distance between satellite $A$ and $B$ at epoch $t_0$; $\Delta\rho_A^B$ and $\Delta\rho_B^A$ denote the differences of the satellite distance and clock bias between the real observation epoch (i.e., $t_A$ and $t_B$) and the naturalization epoch $t_0$. They are defined as follows:

$$\Delta\rho_A^B = \rho_A^B(t_0) - \rho_A^B(t_A + \Delta t_{AB}) + \delta_B(t_0) - \delta_A(t_0) - [\delta_B(t_A + \Delta t_{AB}) - \delta_A(t_A)] \tag{6}$$

$$\Delta\rho_B^A = \rho_B^A(t_0) - \rho_B^A(t_B + \Delta t_{BA}) + \delta_A(t_0) - \delta_B(t_0) - [\delta_A(t_B + \Delta t_{BA}) - \delta_B(t_B)] \tag{7}$$

According to the analysis of BDS-3 ISL data, using broadcast orbits and clock biases to compute $\Delta\rho_A^B$ and $\Delta\rho_B^A$, the errors are less than 1.0 cm [24].

After the above data processing procedure, a clock-free observation can be formed by the sum of the dual measurements, which is:

$$\widetilde{\rho}_{AB}(t_0) = \frac{\widetilde{\rho}_A^B(t_0) + \widetilde{\rho}_B^A(t_0)}{2} = \rho_{AB}(t_0) + \frac{\left(l_A^R - l_A^S + l_B^R - l_B^S\right)}{2} + \frac{\varepsilon_{AB} + \varepsilon_{BA}}{2} \tag{8}$$

From the above equation, we can see the final observation $\widetilde{\rho}_{AB}(t_0)$ does not have clock bias parameters, which are used in the POD process [37].

## 3. Simulation and Method

### 3.1. LEO Constellation Design

Since most of the LEO constellations are still in the preparatory stage, three kinds of LEO constellations are designed for the simulation of ISLs. As for satellite numbers, 60, 120, and 192 are selected for comprehensive analysis. It is noted that the 60-satellite design is also adopted by the HongYan constellation program. All of these LEO satellites are evenly distributed in 10 or 12 equally spaced orbital planes with an altitude of 1000 km. In the design of orbit type, polar orbit is chosen. For simplicity, the pseudo random noise (PRN) codes of these constellations are arranged from 1 to 60, 1 to 120, 1 to 192, respectively. Detailed parameters of the designed LEO constellations are shown in Table 1 and the configurations are displayed in Figure 2.

**Table 1.** Detailed orbital characteristics of the designed low Earth orbit (LEO) constellations.

| Orbit | LEO | LEO | LEO |
|---|---|---|---|
| Satellite number | 60 | 120 | 192 |
| Constellation | 10 planes | 10 planes | 12 planes |
| Orbit type | Polar | Polar | Polar |
| Inclination [deg] | 90 | 90 | 90 |
| Altitude [km] | 1000 | 1000 | 1000 |

### 3.2. ISL Topology Design

The topology of the ISL shows how the link is connected, which is closely related to the geometric structure of the range-measuring network and has a great impact on the results of orbit determination. Chinese BDS-3 uses such a link topology: as long as a pair of satellites are visible to each other, a link is established [29]. In the following text, this link topology is recorded as "all-connected". Obviously, "all-connected" link topology can get the maximum numbers of links in a constellation, which helps to enhance real-time communications between satellites thus easily controlling the operation of the entire constellation. On the other hand, this kind of link topology can also achieve the optimal geometry of the inter-satellite ranging network, and it is expected to obtain high-precision orbit determination results. Although "all-connected" link topology has the abovementioned advantages, it will result in high construction price and large calculation costs. For the comprehensive consideration of workload and POD effectiveness, some scholars have proposed another so-called "4-connected" link topology in which each satellite has two intra-orbit links with neighbors fore and aft in the same orbital plane and two inter-orbit links with satellites in adjacent planes to either side [38]. It is worth noting that the "4-connected" scheme has been adopted by the Iridium constellation [39]. Considering that the "all-connected" link topology can achieve higher orbit determination accuracy and better demonstrate the effect of ISL POD, we mainly use the "all-connected" scheme for simulation experiments and the "4-connected" scheme is also adopted for comparison.

### 3.3. ISL Observation Simulation

Since there is no actual measured data of the LEO ISLs, we simulate the ISL observations of three constellations mentioned above. Simulation of ISL observations usually adopts the approach that adding Gaussian noises to the actual calculated inter-satellite ranges [25,40,41], which can be formulated as follows:

$$S_{ij} = \sqrt{\left(x_j - x_i\right)^2 + \left(y_j - y_i\right)^2 + \left(z_j - z_i\right)^2} \tag{9}$$

$$S_{ij}' = S_{ij} + \sigma_{ij} \tag{10}$$

where $S_{ij}$ is the real inter-satellite range between satellite $i$ and satellite $j$ which is calculated by the simulated 3D coordinates $(x_i, y_i, z_i, x_j, y_j, z_j)$ of each satellite in the Earth-centered inertial (ECI) frame; $S_{ij}'$ denotes the simulated ISL measurements between two satellites computed by the sum of $S_{ij}$ and Gaussian noise $\sigma_{ij}$. In this study, the coordinates $(x, y, z)$ of the polar-orbiting satellites in ECI frame are calculated as follow:

$$x = R cos f cos \Omega \quad y = R cos f sin \Omega \quad z = R sin f \tag{11}$$

where $R$ denotes the spatial distance from the satellite to the center of the Earth; $f$ denotes the true anomaly and $\Omega$ denotes the right ascension of the ascending node (RAAN) of the satellite. $R$, $f$, and $\Omega$ can be obtained according to the configuration of the designed constellation.

In accordance with the studies of predecessors, the measurement noise of BDS-3 ISL is confirmed to be less than 10 cm [29]. Because of the more complex space environment in which the LEO constellation operates, we mainly adopt numerically larger 20 cm-level noise scheme in this study.

For comparison, 10 and 40 cm-level noise scheme are also used to verify the impact of link accuracy on orbit determination.

### 3.4. POD Methods

For any ISL observation in the constellation, it can form a formula as follow:

$$v_{ij} = -\frac{\Delta x_{ij}}{\widetilde{S}_{ij}}\hat{x}_i - \frac{\Delta y_{ij}}{\widetilde{S}_{ij}}\hat{y}_i - \frac{\Delta z_{ij}}{\widetilde{S}_{ij}}\hat{z}_i + \frac{\Delta x_{ij}}{\widetilde{S}_{ij}}\hat{x}_j + \frac{\Delta y_{ij}}{\widetilde{S}_{ij}}\hat{y}_j + \frac{\Delta z_{ij}}{\widetilde{S}_{ij}}\hat{z}_j - \widetilde{l}_{ij} + \varepsilon_{ij} \tag{12}$$

where $v_{ij}$ is the residual of ISL measurement; $(\hat{x}_i, \hat{y}_i, \hat{z}_i)$ and $(\hat{x}_j, \hat{y}_j, \hat{z}_j)$ are the corrections of satellite coordinates; $\Delta x_{ij}$, $\Delta y_{ij}$, and $\Delta z_{ij}$ are coordinate differences calculated by rough starting orbital data; $\widetilde{l}_{ij}$ denotes the difference between link observations and $S_{ij}$; $\varepsilon_{ij}$ is the measurement noise.

As long as the numbers of link observations in the constellation are greater than the numbers of satellite coordinates to be sought, the orbits of the entire constellation can be precisely determined under high-precision inter-satellite geometric constraints. According to the principle of the whole network adjustment, precise orbits are calculated as follow:

$$\hat{x} = \left(B^T P B\right)^{-1} B P l \tag{13}$$

where $\hat{x}$ is corrections of coordinates of satellites in ECI; $B$, $P$, and $l$ (which is formed by $\widetilde{l}_{ij}$ in each link) denote the coefficient matrix, weight matrix, and residual matrix, respectively.

In the absence of ground stations or anchor stations to provide a spatial reference, relying solely on inter-satellite orbit determination will result in overall constellation rotation [25]. In essence, the above coordinate solving process is similar to the rank-deficient free network adjustment. Without considering the scale information of the free network, an n-dimensional ranging network needs to provide $n$ position references and $\frac{n(n-1)}{2}$ direction references. Since the inter-satellite ranging network is a 3D mesh shape, the rank deficit is 6. The spatial reference of the entire constellation can be determined by simply giving the coordinates of the two LEO satellites. In our study, we simulate the coordinates of the two LEO satellites determined by the navigation satellite data to solve the rank deficit problem in the network adjustment.

In addition to the abovementioned method, it is also possible to use the center of gravity reference to perform ISL POD without relying on satellite coordinate data. Assuming that the sum of the corrections of all satellite coordinates in the entire constellation (i.e., the sum of vector $\hat{x}$) is zero, the auxiliary matrix $S$ is introduced, and it satisfies the following formula:

$$S^T \hat{x} = 0 \tag{14}$$

In an ISL network, the specific form of the $S$ matrix is detailed in Fan et al. [42]. Then, precise orbits determined using center of gravity reference can be calculated as:

$$\hat{x} = \left(B^T P B + S S^T\right)^{-1} B P l \tag{15}$$

## 4. Results and Analyses

In this section, we mainly focus on the demonstration of POD results with ISLs and relevant analysis is conducted. The whole section is divided into three parts. In the first part, experimental results based on two link topologies are demonstrated. Then, POD results with center of gravity reference are shown and we concentrate on analyzing the impact of link ranging accuracy and the numbers of links on the performance of POD. Finally, POD results with reference provided by navigation satellites are shown to demonstrate that the high precision can be achieved by ISL POD.

### 4.1. Test with Different Link Topologies

Two ISL link topologies, named "all-connected" and "4-connected", have been introduced in the Section 3.2. It is obvious that link topology has a great impact on the orbit determination performance. For test, the 60-LEO constellation detailed in the Table 1 is selected, then the ISLs based on both link topologies are simulated. It is noted that average numbers of links per satellite are 16.7 with "all-connected" link topology. For the sake of simplicity, we take a rough orbital solution for each satellite in the constellation. Assuming that the coordinates of the satellites linked to each satellite are known, the coordinates of each satellite are determined by precise positioning. Average orbital errors are shown in the Table 2. It can be concluded that except for the errors along the z-axis, orbital errors are much smaller with the "all-connected" link topology than with the "4-connected" link topology. Numerical analysis shows that when adopting "all-connected" link topology, there is 38.4% reduction of average 3D orbital errors compared to adopting "4-connected" link topology.

**Table 2.** Average orbital errors of 60-LEO constellation with different link topologies.

| Errors | $\Delta x$/m | $\Delta y$/m | $\Delta z$/m | 3D/m |
|---|---|---|---|---|
| 4-connected | 0.095 | 0.045 | 0.038 | 0.112 |
| all-connected | 0.052 | 0.002 | 0.045 | 0.069 |

According to the above brief analysis, "all-connected" link topology shows better orbit determination performance than "4-connected" link topology. In order to better demonstrate the effectiveness of ISL POD, "all-connected" link topology is used in the following study.

### 4.2. POD with Center of Gravity Reference

Using ISL measurements, POD with center of gravity reference mentioned in Section 3.4 can be achieved. In order to study the factors affecting the POD results, we carry out comparative experiments with several sets of satellite numbers and different observation accuracy. Because the numbers of satellites are very large in our experiments, for the sake of convenience the satellites are displayed at intervals 3, 6, 9 of PRN number for 60-, 120-, 192-LEO constellations, respectively. This means there are 20, 20, 22 satellites to be shown in the 60-, 120-, 192-LEO constellations, respectively.

First, in order to verify the impact of link ranging accuracy on the orbit determination, we select 60-LEO constellation detailed in the Table 1 for test. Experiments are carried out with 40, 20, and 10 cm ISL ranging accuracy and the orbital errors in the radial, along-track, cross-track, and 3D directions are shown in the Figures 3–6. From these figures, it can be clearly concluded that higher ranging accuracy leads to lower orbital errors in every direction. It is because that observation errors which compose *l* in Equation (15) are smaller during adjustment process. Average orbital errors of 60-LEO constellation after POD with 40, 20, and 10 cm ranging accuracy are shown in the Table 3. When the ranging accuracy is 40 cm, POD result is poor with average 3D error of 0.358 m but it turns to be much better when the ranging accuracy is 20 or 10 cm. It can be calculated that compared to the ranging accuracy of 40 cm, the 20 and 10 cm schemes bring about 76.0 and 93.9% 3D orbital error reductions, respectively.

**Table 3.** Average orbital errors of 60-LEO constellation under different ranging accuracies.

| Direction | Ranging Accuracy | | |
|---|---|---|---|
| | 40 cm | 20 cm | 10 cm |
| Radial [m] | 0.179 | 0.043 | 0.011 |
| Along-track [m] | 0.084 | 0.020 | 0.005 |
| Cross-track [m] | 0.272 | 0.065 | 0.016 |
| 3D [m] | 0.358 | 0.086 | 0.022 |

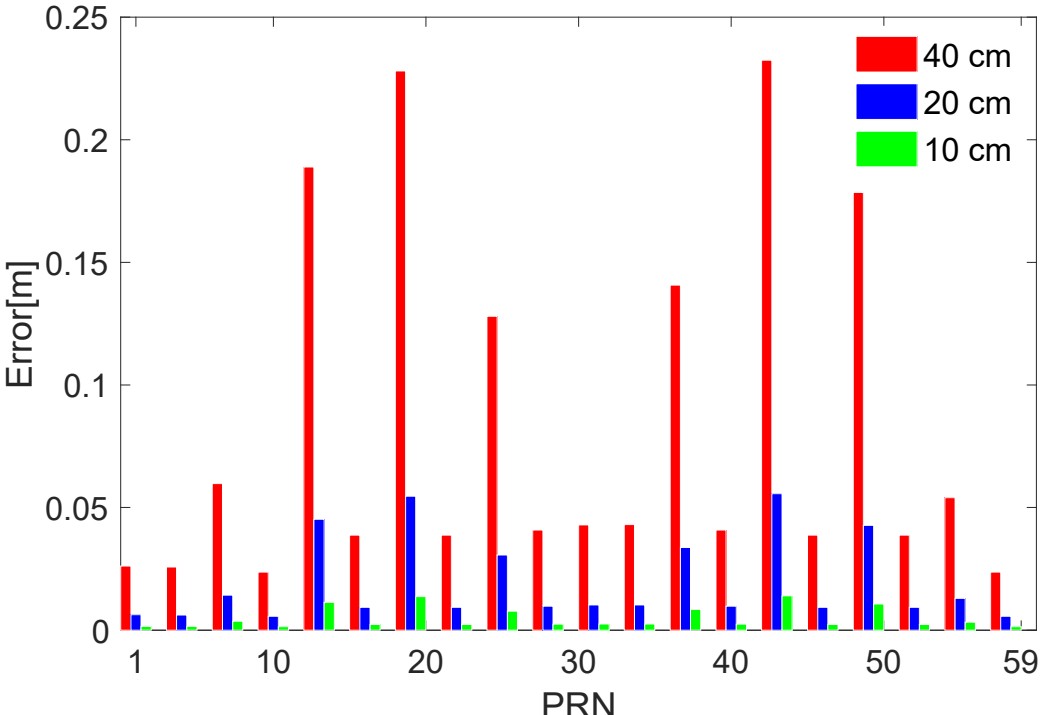

**Figure 3.** 60-LEO constellation orbital errors in the radial direction under different ranging accuracies.

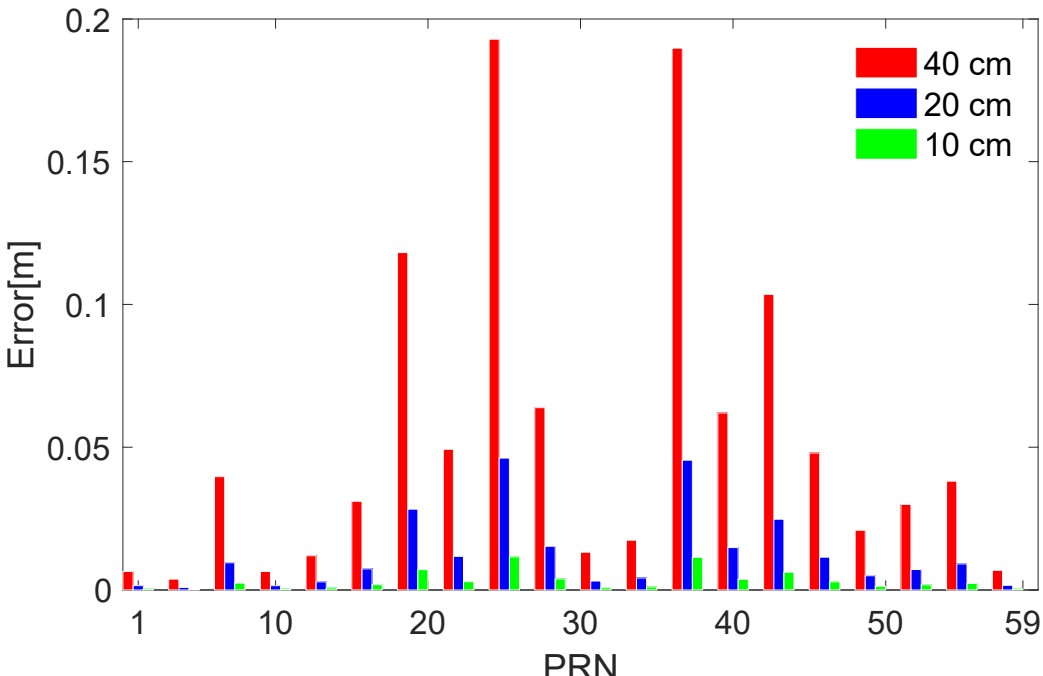

**Figure 4.** 60-LEO constellation orbital errors in the along-track direction under different ranging accuracies.

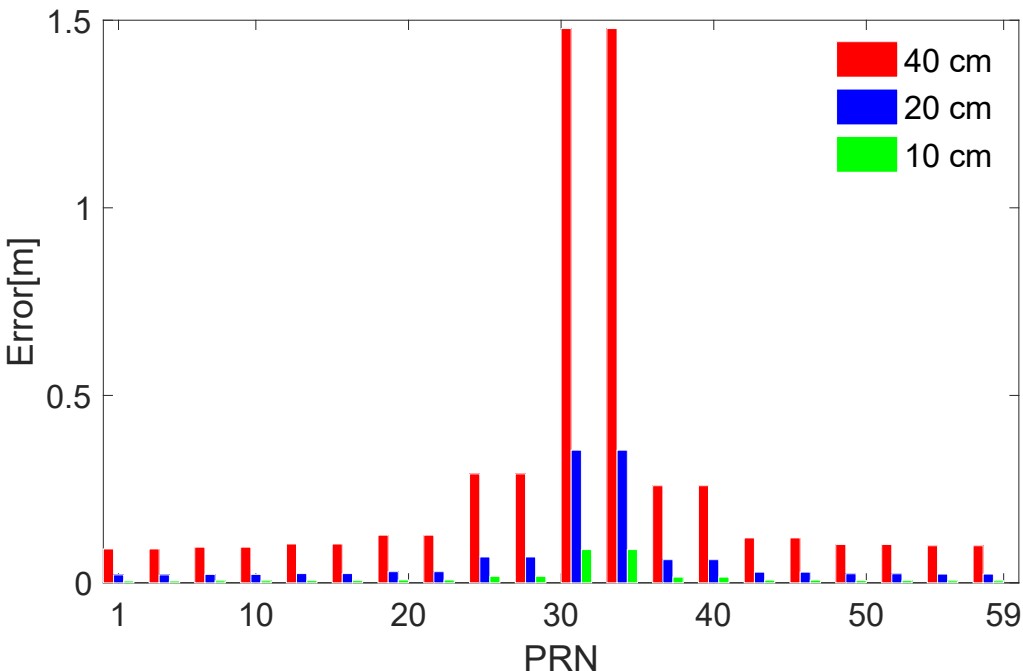

**Figure 5.** 60-LEO constellation orbital errors in the cross-track direction under different ranging accuracies.

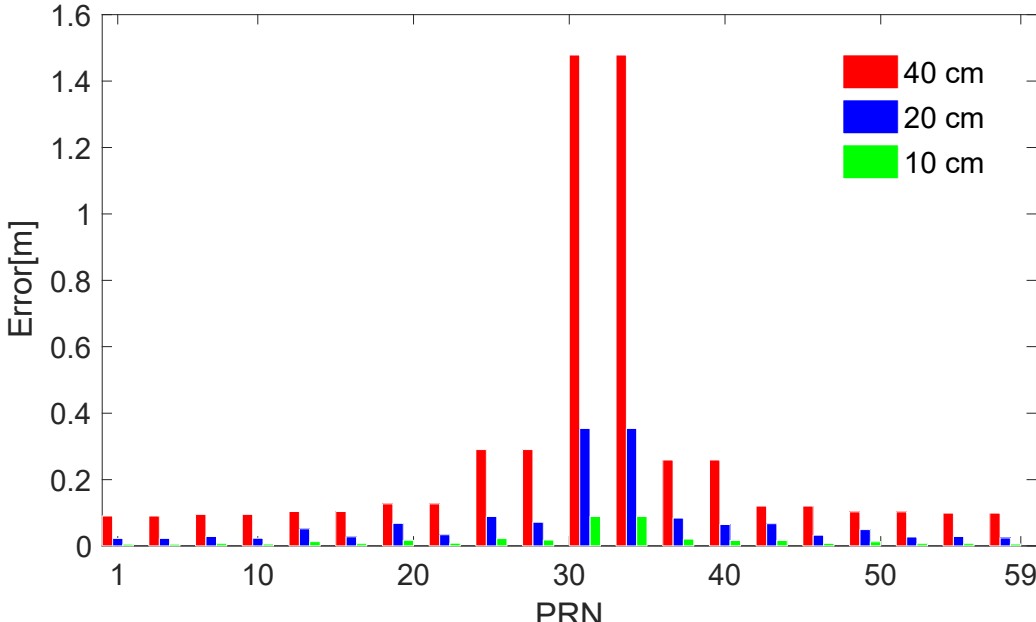

**Figure 6.** 60-LEO constellation 3D orbital errors under different ranging accuracies.

Then, several experiments are carried out to verify the impact of link numbers on the orbit determination. It is obvious that the more satellites in a constellation, the more ISLs will be formed, so we select 60-, 120-, and 192-LEO constellations detailed in the Table 1 for further study and the 20 cm ISL ranging accuracy is chosen for all constellations. The orbital errors of these constellations are shown in the Figures 7–9. Figure 10 shows the average orbital errors in different directions of these constellations. Average numbers of links per satellite in these constellations are 16.7, 33.0, and 51.5, respectively. Obviously, the more satellites, the more links there are, which contributes to the smaller orbital errors in all directions. It is because more link observations, which means the number of rows in matrix *l* in Equation (15) are larger, brings stronger geometric constraints to the coordinates of the

satellites. The average 3D orbital error is 0.086 m for 60-LEO constellation while the errors are 0.030 and 0.009 m for 120- and 192-LEO constellations with 65.1 and 89.5% reductions, respectively.

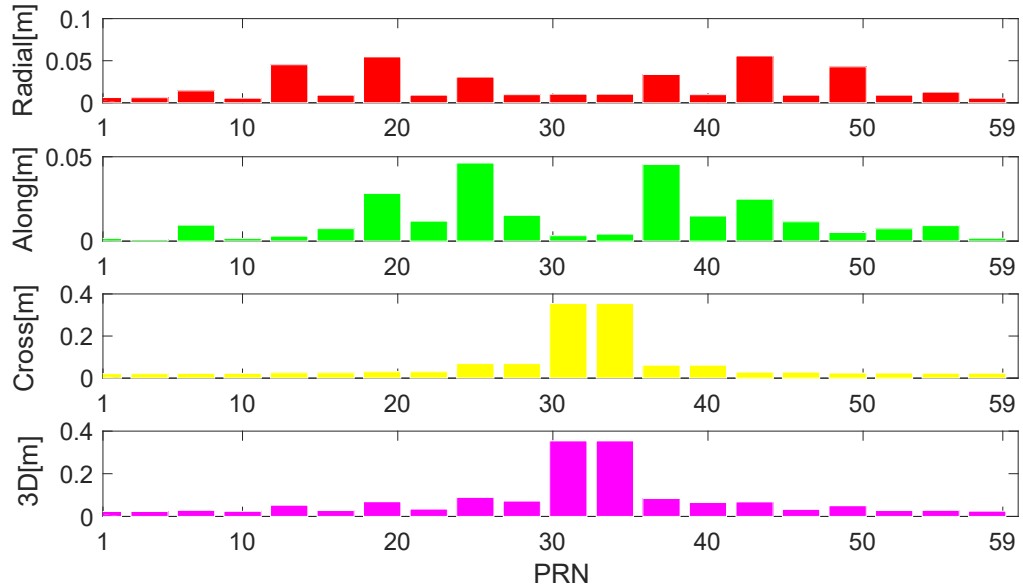

**Figure 7.** 60-LEO constellation orbital errors in radial, along-track, cross-track, and 3D under 20 cm ranging accuracy, in which satellite of PRN 1, 4, 7, . . . , 58 (divided at interval 3 of PRN number) are displayed.

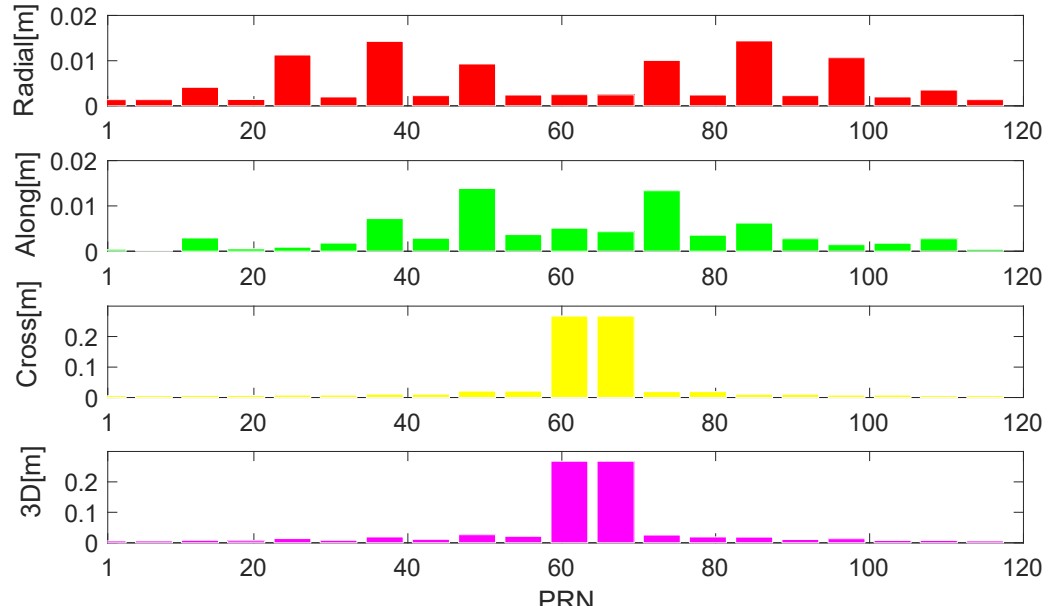

**Figure 8.** 120-LEO constellation orbital errors in radial, along-track, cross-track and 3D under 20 cm ranging accuracy, in which satellite of PRN 1, 7, 13, . . . , 115 (divided at interval 6 of PRN number) are displayed.

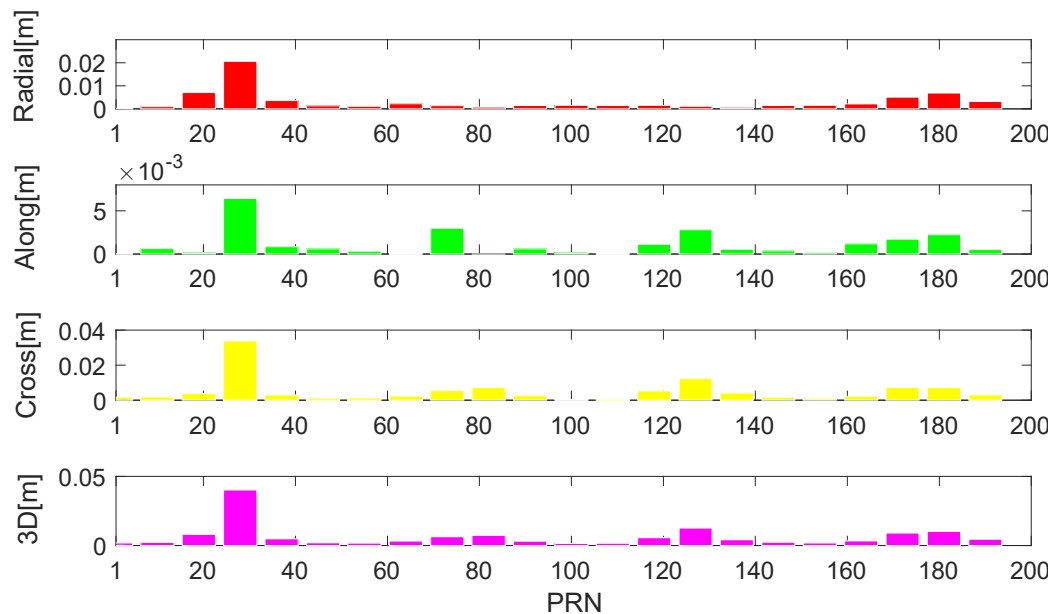

**Figure 9.** 192-LEO constellation orbital errors in radial, along-track, cross-track and 3D under 20 cm ranging accuracy, in which satellite of PRN 1, 10, 19, . . . , 190 (divided at interval 9 of PRN number) are displayed.

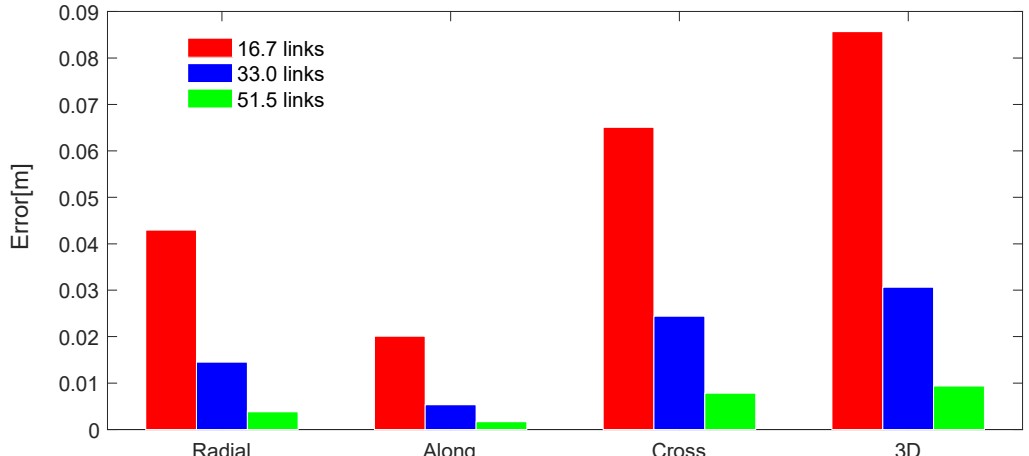

**Figure 10.** Average orbital errors under 20 cm ranging accuracy of the 60-LEO (**red**), 120-LEO (**blue**) and 192-LEO (**green**) constellations.

### 4.3. POD with Reference Provided by Navigation Satellites

According to the above analysis, POD with ISLs gets bad performance in the case of center of gravity reference when link ranging accuracy is poor and numbers of links are small. When the link accuracy is 40 cm, average 3D orbital errors of 60-LEO constellation are 0.358 m, which can only meet the demand of autonomous navigation. Compared to a center of gravity reference, a spatial reference which is decided using GNSS data from onboard LEO receiver is more common in use and it is confirmed to have better POD potential. In our study, two satellites in each constellation are considered as "reference satellites" and their 3D coordinates, which are assumed to be decided by navigation satellites, are simulated by adding 3 cm noises to the x, y, and z components of their real coordinates.

The accuracy of POD can be improved with constraint of orbital parameters [43]. Previous studies have shown that the predicted accuracy of RAAN is very high for the MEO and GEO satellites [25], thus the orbit determination accuracy can be effectively improved by using the RAAN predicted value

to impose constraints to the observation functions. Though the orbital parameters of the LEO satellites are more changeable because of relatively low orbit altitudes, RAAN values of LEO satellites are still stable within a few hours [44,45]. Therefore, it is feasible to constrain RAAN to improve the orbit determination accuracy of LEO satellites during short periods.

Since more numbers of links bring better POD performance as mentioned above, 120- and 192-LEO constellations are selected for test and the link ranging accuracies are both 20 cm. RAANs of each orbit in both constellations are regarded as known values when computing the link ranging residuals which form the matrix *l* in the network adjustment process as shown in Equation (13). Average link ranging residuals and total numbers of links of both constellations are displayed in Table 4. It can be calculated that when using known RAAN values as constraints, average ranging residuals get 68.0% reduction for 120-LEO constellation and 48.6% for 192-LEO constellation, which means the elements that make up the matrix *l* in Equation (13) have smaller absolute values. Then, the orbital errors of experimental constellations are shown in Figures 11 and 12. It can be clearly seen from the figures in both constellations, orbital errors of all the satellites displayed are very small. For further detailed analysis, average orbital errors of both constellations are shown in the Table 5. We can conclude from the table that average orbital errors of both constellations in every direction are less than 1 cm and 192-LEO constellation achieves slightly better POD performance than 120-LEO constellation. Obviously, with RAAN as the constraint, POD by using the reference provided by navigation satellites reaches an extremely high precision. It could have resulted from the strong constellation geometry with the constraints of RAANs and the given reference.

**Table 4.** Average link ranging residuals and total numbers of links of 120- and 192-LEO constellation.

| Constellation | Ranging Residuals/m | | Numbers of Links |
| --- | --- | --- | --- |
| | **Without RAAN Constraint** | **With RAAN Constraint** | |
| 120-LEO | 0.344 | 0.110 | 1982 |
| 192-LEO | 0.625 | 0.321 | 4942 |

**Figure 11.** 120-LEO constellation orbital errors in radial, along-track, cross-track, and 3D under 20 cm ranging accuracy, in which satellite of PRN 1, 7, 13, . . . , 115 (divided at interval 6 of PRN number) are displayed.

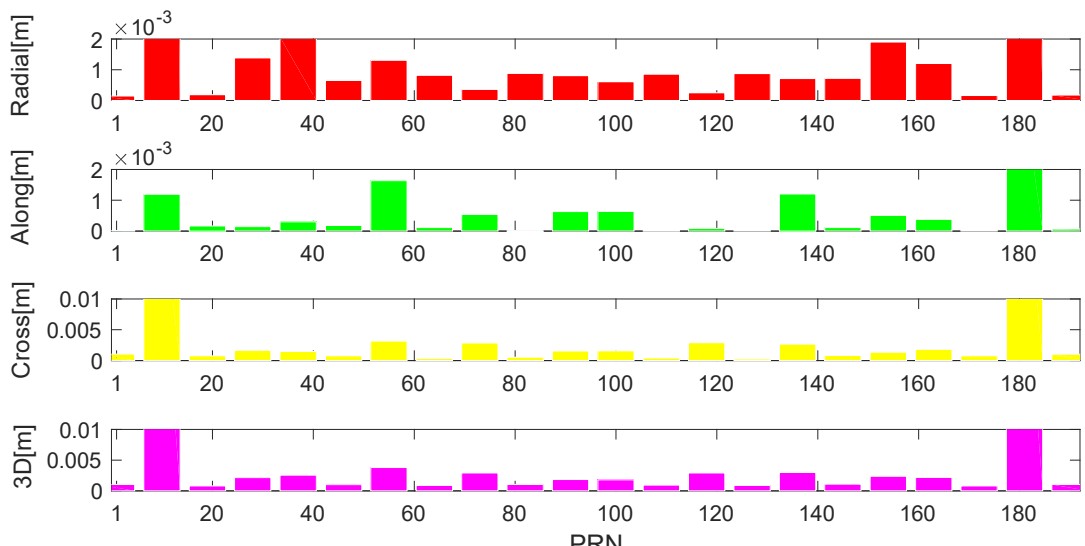

**Figure 12.** 192-LEO constellation orbital errors in radial, along-track, cross-track and 3D under 20 cm ranging accuracy, in which satellite of PRN 1, 10, 19, ..., 190 (divided at interval 9 of PRN number) are displayed.

**Table 5.** Average orbital errors of 60- and 120-LEO constellations with references provided by navigation satellites.

| Constellation | Orbital Errors/m | | | |
|---|---|---|---|---|
| | Radial | Along-Track | Cross-Track | 3D |
| 120-LEO | 0.005 | 0.002 | 0.008 | 0.010 |
| 192-LEO | 0.003 | 0.001 | 0.005 | 0.006 |

## 5. Discussions

As a high-precision ranging method between satellites, ISLs have been proven by many scholars to be used in the process of POD. Past researches focus on analyzing POD performance of MEO, GEO, and IGSO navigation satellites with ISLs. For the first time, we have made a comprehensive assessment of the POD performance of LEO satellites with ISLs. The results demonstrate the accuracy is considerable because of strong geometric constraint from ISL observations. Moreover, according to the numerical analysis, better link ranging accuracy and more numbers of links get better POD accuracy. This is because there are more observations and smaller observation errors during the adjustment process. Our study shows that future application of ISLs in the POD for LEO satellites is possible.

Limited by the absence of real LEO onboard data, all our work is based on simulation. We only simulate dual-way observations which are used in the POD process. This simulation strategy circumvents the refinement preprocessing of raw observation data so the POD process is greatly simplified compared to the actual situation. Moreover, in the navigation constellations, the establishment of ISLs usually adopts the principle of time division multiple access (e.g., BDS-3). It means every satellite establishes links with other satellites according to the designed timeslot but at an exact time there is only one link per satellite. For simplicity, we assume that at one point in time, all satellites can be linked to multiple satellites to avoid epoch naturalization. For the above reasons, our simulated data is somewhat inconsistent with real data. Therefore, the high-precision orbit determination results in our study have not yet been tested in practical applications.

The method that we used for POD is a simple kinematic orbit determination based on the whole network adjustment, but in common use reduced-dynamic orbit determination which take force models into consideration is mainly adopted for the LEO satellites. It is a pity that we can't find a

proper force model for simulated constellation so we refer to previous studies using a simpler orbit determination method. In the following study, we will improve our means of orbit determination.

In general, we have carried out some preliminary research by focusing on demonstrating the feasibility of carrying out POD process for LEO satellites using ISLs. Subsequent study relies on the refinement of data processing methods and the acquisition of actual ISL data.

## 6. Conclusions

We carry out research on LEO satellite POD using ISLs for the first time with designed constellations and simulated ISL observations. The experimental results evidence that POD with ISLs can achieve high precision for LEO satellites.

There are two commonly used ISL link topologies: "all-connected" link topology with theoretically maximum numbers of links and "4-connected" link topology with constant four links per satellite. Rough orbit determination tests show the average 3D orbital errors are 0.112 m with "4-connected" link topology and 0.069 m with "all-connected" link topology in the 60-LEO constellation. POD performance can be better demonstrated with "all-connected" link topology.

POD with center of gravity reference gets bad result when the ranging accuracy is poor and numbers of links are smaller. In the 60-LEO constellation, average 3D orbital error is 0.358 m with 40 cm ranging accuracy. The 20 cm ranging accuracy and the 10 cm ranging accuracy bring about 76.0% and 93.9% 3D orbital errors reductions compared to 40 cm ranging accuracy, respectively. Under the same 20 cm ranging accuracy, the average 3D orbital errors of 120- and 192-LEO constellations are 0.030 and 0.009 m, with 65.1% and 89.5% reductions compared to the 60-LEO constellation, respectively. Average 3D orbital errors are less than 10 cm for all constellations with 20 cm ranging accuracy. POD results with the center of gravity reference show the better link ranging accuracy and the more numbers of links bring the better orbit determination performance.

With RAAN as the constraint, POD by using the reference provided by navigation satellites reaches an extremely high precision. Under 20 cm link accuracy, average orbital errors of 120-LEO constellation are 0.005 m in the radial, 0.002 m in the along-track, and 0.008 m in the cross-track and the 3D orbital error is 0.010 m. Average orbital errors of 192-LEO constellation are slightly better than 120-LEO constellation, with 0.003 m in the radial, 0.001 m in the along-track, 0.005 m in the cross-track, and 0.006 m in the 3D.

In summary, high-precision dual-way ranging based on ISLs enables POD for LEO satellites. Generally, with better link ranging accuracy, more numbers of links, and better POD performance can be achieved.

**Author Contributions:** All authors contributed to the writing of this manuscript. X.L. (Xingxing Li), Z.J. and F.M. conceived and designed the experiments. Z.J. and F.M. processed the data, drew the pictures, and investigated the results. H.L., Y.Y. and X.L. (Xin Li) validated the data.

**Funding:** This study is financially supported by the National Natural Science Foundation of China (Grant No. 41774030), the Hubei Province Natural Science Foundation of China (Grant No. 2018CFA081), the Wuhan Science and Technology Project (Grant No. 2018010401011270), and the National Youth Thousand Talents Program.

**Acknowledgments:** We are very grateful to the reviewers and the editors for their helpful remarks for improving the manuscript. Meanwhile, we would also like to express our gratitude to Yanling Chen from Shanghai Astronomical Observatory, Chinese Academy of Sciences. She enthusiastically provided guidance for our study, bringing great help to the early research. In addition, our codes for the research have adopted several functions from open source library RTKLIB. We sincerely thank the code founder Tomoji Takasu as well as other people who have contributed to improving RTKLIB.

**Conflicts of Interest:** The authors declare no conflict of interest.

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
