# Peer review of "LEO Precise Orbit Determination with Inter-satellite Links"

_remotesensing, doi:10.3390/rs11182117_

Round 1
Reviewer 1 Report
The paper shows a nice simulation study of how LEO satellites orbits can be computed and improved. The language is mostly clear and the theory, methods and results are presented in an appropriate format. Some minor remarks:
The abbreviations are not explained: GPS, GLONASS, ADVISE, BDS
In the introduction is mentioned that there are “some scholars”, who have done something. I would prefer references.
In explanation of formula (2) c is missing, I assume speed of light.
In formulas 2-5 the variables with tilde-sign are not explained. I assume the first ones are dual pseudo-code measurements and the latter ones the pure dual pseudo-code measurements, but add the symbols in to the text.
Is the l in (12) the same as in (13)?
In figures 2-4 (and all other histogram plots) it is disturbing that the histograms reach the top of the plot. It’s impossible to know if the bar stops there of if it’s just cut there. Please increase the scale.
Figures 7 and 8 as well as 10 and 11 have identical captions. Please add little more details to justify the figures.
The references are not properly presented. There are years missing, as well as crucial information on publication, e.g. “International Conference on Localization & GNSS. IEEE” is not enough. Some of the references seem to be theses of some kind, but I cannot find them, are they in Chinese? If so, it should be clearly stated.
Some words (find with CRTL+F) that I find inappropriate/unsuitable for scientific text:
war (how common is war as a communication error source?) “due to any reason/different reasons”?
descripted (described?)
refined (defined?)
truthful data (real data? the whole sentence is a bit cryptic)
large calculation amounts (calculation costs?)
gauss noise = Gaussian noise!
Reviewer 2 Report
The manuscript aims to present the results of the orbit determination of large constellations carried out by inter-satellite-links (ISL). Topic is certainly of high interest, and results (while in some way expected) can be useful and significant to readers to provide them with an idea about attainable accuracy. Graphics are also good quality. However, in the opinion of this reviewer, the manuscript requires some attention:
- first of all, the introduction, and in some way also the text, is not really clear about LEO and higher orbit case (by the way, the altitude of the "higher" cases is not even specified), as well as about ground vs. in-orbit measurements cases. A suggestion could be to describe one case, and then the second one (for each of the two topics), without continuously mixing them. Moreover, the inter-satellite-link technology, that is the basis for this application, should be at least briefly reported (which frequency band is used, which hardware is required and so on: the only rather vague comments are about the nature (microwave or laser, page 2) and the pseudorandom coding (page 4). In addition, it could be useful to include in the introduction that inter-satellite measurements could be done in different ways, with different degrees of accuracy: a recall to papers like "Moderate Accuracy Relative Navigation in Formation Flying by Filtered Radio Measurements" (IEEE Aerospace Conference Proceedings, 2015) and, on the more advanced side, “Intersatellite laser ranging instrument for the GRACE follow-on mission”, Journal of Geodesy, 2012) could be interesting for readers.
- most important: results are clear, while the description of the fundamental relations of the process can be improved, i.e. better explained: it is the case of Eq.1 and (partly) of Eqs.5-8.
Minor corrections:
introduction, 3rd line: "method of orbit observation" (?): it could be "the measurement technique exploited/adopted" or the "nature of measurements" ?
page 3: "will complete the networking" should better be "will complete the deployment";
page 4: "to be constants" should be "to be constant";
page 5, 3rd line: what is HongYangconstellation program?
ref. 11 is not complete, please add information (at least the year) to allow readers to locate this resource. The year should be added also to references 4 and 6;
ref.13 "Russia" instead of Rusia;
ref. 20 could be completed by the access date (like January 2019..)
